# Coherent dynamics of strongly interacting electronic spin defects in hexagonal boron nitride

Ruotian Gong[1], Guanghui He[1], Xingyu Gao[2], Peng Ju[2], Zhongyuan Liu[1], Bingtian Ye [3,4], Erik A. Henriksen [1,5], Tongcang Li [2,6] & Chong Zu [1,5] ✉

Optically active spin defects in van der Waals materials are promising platforms for modern quantum technologies. Here we investigate the coherent dynamics of strongly interacting ensembles of negatively charged boron-vacancy ($V_B^-$) centers in hexagonal boron nitride (hBN) with varying defect density. By employing advanced dynamical decoupling sequences to selectively isolate different dephasing sources, we observe more than 5-fold improvement in the measured coherence times across all hBN samples. Crucially, we identify that the many-body interaction within the $V_B^-$ ensemble plays a substantial role in the coherent dynamics, which is then used to directly estimate the concentration of $V_B^-$. We find that at high ion implantation dosage, only a small portion of the created boron vacancy defects are in the desired negatively charged state. Finally, we investigate the spin response of $V_B^-$ to the local charged defects induced electric field signals, and estimate its ground state transverse electric field susceptibility. Our results provide new insights on the spin and charge properties of $V_B^-$, which are important for future use of defects in hBN as quantum sensors and simulators.

Solid-state point defects with optically addressable electronic spin states have become some of the most fertile playgrounds for new quantum technologies[1-18]. Significant recent progress has been made in creation and control of such spin-active quantum emitters in atomic-thin van der Waals materials. The two-dimensional (2D) nature of the host materials can enable seamless integration with heterogeneous, optoelectronic, and nanophotoic devices, providing a pathway to investigating light-matter interactions at the nanoscale[19-22].

From a wide range of contestant spin defects in 2D materials, the negatively charged boron vacancy center, $V_B^-$, in hexagonal boron nitride (hBN) has particularly attracted substantial research interest in the past few years[23-32]. Importantly, it has been demonstrated that the spin degree of freedom of $V_B^-$ can be optically initialized and readout, as well as coherently manipulated at room temperature. Compared to conventional spin qubits in three-dimensional materials, such as nitrogen-vacancy (NV) center in diamond, $V_B^-$ features several unique advantages in quantum sensing and simulation.

From the perspective of quantum sensing, the atomically-thin structure of hBN can allow the $V_B^-$ sensor to be positioned in close proximity with the target materials, facilitating the imaging of interfacial phenomena with unprecedented spatial resolution and sensitivity[25,33–35]. Moreover, since hBN has been widely employed as the encapsulation and gating dielectric material in 2D heterostructure devices, introducing the embedded $V_B^-$ sensors does not require any additional complexity in the fabrication process[36–40]. On the quantum simulation front, the ability to prepare and control strongly interacting, two-dimensional spin ensembles opens the door to exploring a number of intriguing many-body quantum phenomena[41–43].

[1]Department of Physics, Washington University, St. Louis, MO 63130, USA. [2]Department of Physics and Astronomy, Purdue University, West Lafayette, IN 47907, USA. [3]Department of Physics, Harvard University, Cambridge, MA 02138, USA. [4]Department of Physics, University of California, Berkeley, CA 94720, USA. [5]Institute of Materials Science and Engineering, Washington University, St. Louis, MO 63130, USA. [6]Elmore Family School of Electrical and Computer Engineering, Purdue University, West Lafayette, IN 47907, USA. ✉e-mail: zu@wustl.edu

For instance, dipolar interaction in 2D is particularly prominent from the perspective of localization and thermalization, allowing one to experimentally investigate the effect of many-body resonances[44–51].

$V_B^-$ in hBN, like solid-state spin defects in general, suffers from decoherence. To this end, research effort has been devoted to characterizing the coherence time of $V_B^-$. However, the measured spin echo timescale, $T_2^{Echo}$, in several studies varies from tens of nanoseconds to a few microseconds[24,52–54]. This immediately begs the question that where does such discrepancy originate from, and what are the different decoherence mechanisms in dense ensemble of $V_B^-$?

In this letter, we present three main results. First, we introduce a robust differential measurement scheme to reliably characterize the spin coherent dynamics of $V_B^-$ ensemble (Figs. 1 and 2). We observe spin-echo $T_2^{Echo} \approx 70$ ns across three hBN samples with distinct $V_B^-$ densities (created via ion implantation with dosages spanning two orders of magnitude), consistent with the expectation that the spin-echo coherence time is dominated by the Ising coupling to the nearby nuclear spin and dark electronic spin bath[52,55]. By applying a more advanced dynamical decoupling sequence, XY-8, to better isolate $V_B^-$ from the bath spin environment[56–58], we observe substantial extensions in the measured coherent timescales, $T_2^{XY8}$. Interestingly, the extracted $T_2^{XY8}$ decreases with increasing $V_B^-$ density, indicating that the dipolar interaction within the $V_B^-$ ensemble is critical for understanding the coherent dynamics. To further corroborate this, we utilize the DROID pulse sequence to decouple the $V_B^- - V_B^-$ dipolar interaction[59,60], and achieve an additional ~2-fold improvement in the measured coherence time, $T_2^D$. Second, by comparing the experimentally measured $T_2^{XY8}$

and $T_2^D$ to numerical simulations, we directly esimtate the spin density of $V_B^-$ across three hBN samples. We find that the ratio of negatively charged $V_B^-$ to total created boron vacancy defects ($V_B$) decreases significantly with increasing ion implantation dosage (Fig. 3). Third, based on the extracted $V_B^-$ density, we introduce a microscopic model of local charges surrounding a spin defect to account for the observed energy splitting between $|m_s = \pm 1\rangle$ states at zero magnetic field[61,62], and estimate the transverse electric field susceptibility of $V_B^-$ to be around $d_\perp \approx 40$ Hz/(V·cm$^{-1}$) (Fig. 4).

## Results

### Experimental system

To investigate the coherent spin dynamics of $V_B^-$ ensemble at various defect densities, we prepare three hBN samples with different implantation dosages. Specifically, we irradiate hBN flakes (thickness ~100 nm) using 3 keV He$^+$ ion beams with dose densities, $0.30 \pm 0.03$ nm$^{-2}$ (sample S1), $1.1 \pm 0.1$ nm$^{-2}$ (sample S2), and $10 \pm 1$ nm$^{-2}$ (sample S3), respectively, to create $V_B^-$ defects[27,53]. Here error bars on the implantation dosages account for the current fluctuations during the implantation process. We remark that, given an ion implantation dosage, the total created $V_B$ concentration can be estimated via SRIM simulation (see Methods)[63], yet the actual density of the negatively-charged $V_B^-$ has remained unknown.

The $V_B^-$ center has a spin triplet ground state ($|m_s = 0, \pm 1\rangle$), which can be initialized and read out via optical excitation and coherently manipulated using microwave fields[23,30]. In the absence of any external perturbations, the $|m_s = \pm 1\rangle$ states are degenerate and separated from

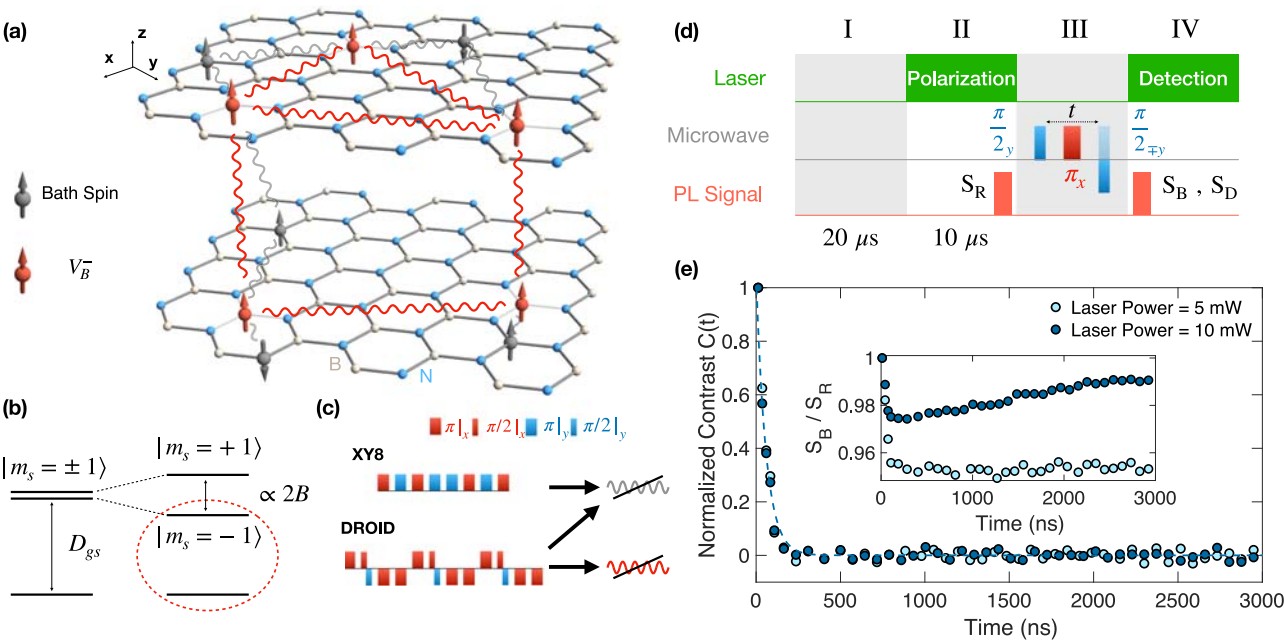

**Fig. 1 | Spin dynamic of $V_B^-$ ensemble. a** Schematic of $V_B^-$ spin ensemble (red spins) inside hBN crystal lattice (Nitrogen--blue; Boron--white). **z** is defined along the c-axis (perpendicular to the lattice plane). **x** and **y** lie in the lattice plane, with **x** oriented along one of the three $V_B^-$ Nitrogen bonds. Two types of decoherence sources are presented here for $V_B^-$ spin ensemble: the Ising coupling (gray wavy lines) to the bath spins (gray), and the dipolar interaction within $V_B^-$ themselves (red wavy lines). **b** Energy level diagram of the defect spin ground-state. In the absence of any external perturbation, the $|m_s = \pm 1\rangle$ states are degenerate and separated by $D_{gs} \approx 3.48$ GHz from the $|m_s = 0\rangle$ state. Under an external magnetic field $B$ along the c-axis of hBN, the degeneracy between $|m_s = \pm 1\rangle$ states is lifted via the Zeeman effect, with a splitting $\propto 2B$. We choose $|m_s = 0\rangle$ and $|m_s = -1\rangle$ states as our two-level system. **c** Experimental pulse sequences for XY-8 (top) and DROID (bottom). The rotations along the positive **x** and **y** axes are plotted above the line, while the

rotations along the negative axes are plotted below the line. **d** Differential measurement sequence for spin echo. I: 20 μs wait time to reach charge state equilibration. II: 10 μs laser pulse to initialize the $V_B^-$ spin to $|m_s = 0\rangle$, with the reference signal, $S_R(t)$, collected at the end of the laser pulse. III: microwave wave pulses for spin echo measurement; for the bright signal, a final $\frac{\pi}{2}$ pulse along the -**y** axis is applied; while for the dark signal, a final $\frac{\pi}{2}$ pulse along the +**y** axis is applied to rotate the spin to an orthogonal state. IV: laser pulse to detect the spin state. **e** Spin echo measurement on sample S3 at two different laser powers. Without differential measurement, the measured signal, $S_B/S_R$ exhibits a laser power dependence which comes from charge relaxation dynamics (inset). Using differential measurement, the measured contrast, $C(t)$, is independent of the laser power. Error bars represent 1 s.d. accounting statistical uncertainties.

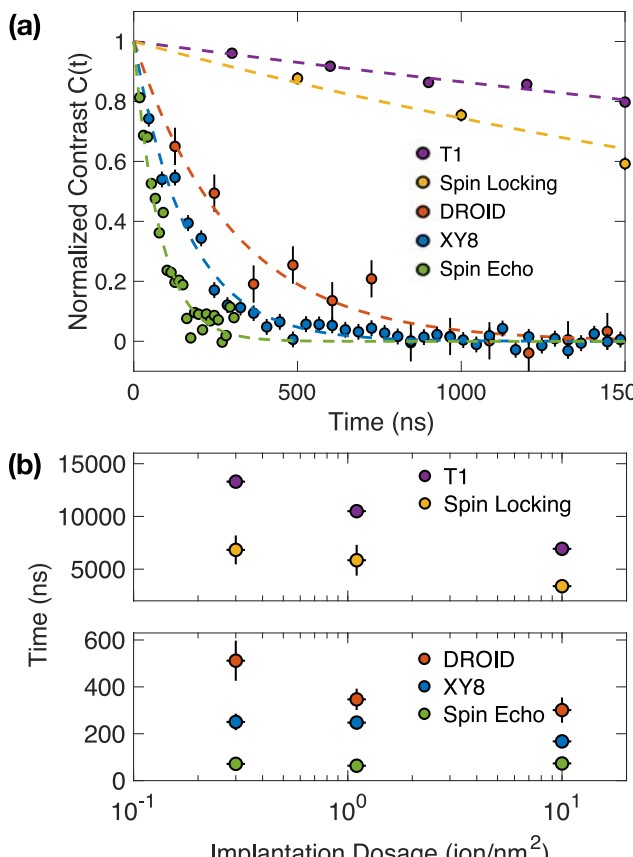

**Fig. 2 | Spin coherent and relaxation dynamics. a** The spin coherent and relaxation timescales measured on sample S3 with the highest ion implantation dosage. Dashed lines are data fitting with single exponential decays. Error bars represent 1 s.d. accounting statistical uncertainties. **b** The extracted coherence timescales $T_2$ and relaxation timescales $T_1$ for the three hBN samples. Error bars in time represents 1 s.d. accounting fitting error, and error bars in implantation dosage represents the current fluctuation induced uncertainties (-10%) during implantation.

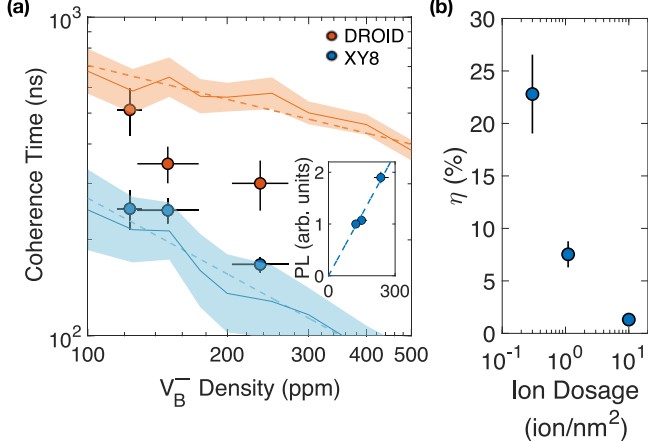

**Fig. 3 | Characterizing $V_B^-$ density. a** Comparison between the experimentally measured and numerically simulated coherent timescales, $T_2$, for DROID and XY-8 pulse sequences. The solid lines show the timescales extracted from simulations with error bars plotted as semi-transparent colored areas. To determine $V_B^-$ densities for the three hBN samples, we minimize the relative squared residuals of $T_2^{XY8}$ and $T_2^D$ between simulations and experiments. Inset: fluorescence counts (PL) versus extracted densities after contrast adjustment (see Methods). Error bars in coherence time account for 1 s.d. of fitting error, and error bars of $V_B^-$ densities represent the range of densities whose residuals lie within 5% (see Methods). **b** The measured $V_B^-$ charge state ratio $\eta = \rho_{V_B^-}/\rho_{V_B}$ for three hBN samples with different ion implantation dosages. Error bars account for both the current fluctuation induced implantation uncertainties and the $V_B^-$ densities uncertainties from (**a**).

$|m_s = 0\rangle$ by $D_{gs} \approx 3.48$ GHz (Fig. 1b). In the experiment, we apply an external magnetic field $B \approx 250$ G along the c-axis of the hBN lattice to separate the $|m_s = \pm 1\rangle$ states via the Zeeman effect and isolate an effective two-level system $|m_s = 0, -1\rangle$. A microwave field is used to coherently manipulate the spin ensemble with a Rabi frequency $\Omega \approx 83$ MHz ($\pi$-pulse length $\tau_\pi = 6$ ns). We note that such a strong Rabi drive is crucial for the high fidelity control of $V_B^-$, as the spin transition is largely broadened by the hyperfine interaction to the nearby nuclear spin bath (see Methods).

### Robust measurement scheme

To reliably probe the spin dynamics of a dense ensemble of $V_B^-$, we introduce a robust differential measurement scheme illustrated in Fig. 1d[64,65]. Specifically, after letting the spin system reach charge state equilibration for 20 $\mu$s without any laser illumination (I), we apply a 10 $\mu$s laser pulse (532 nm) to initialize the spin state of $V_B^-$ (II), followed by the measurement pulse sequences (III). Taking spin echo coherent measurement as an example, we first apply a $\frac{\pi}{2}$-pulse along the **y** axis to prepare the system in a superposition state $\otimes_i \frac{|0\rangle_i + |-1\rangle_i}{\sqrt{2}}$, and then let it evolve for time $t$. A refocusing $\pi$-pulse along the **x** axis at time $t/2$ is used to decouple the spin ensemble from static magnetic noise. A final $\frac{\pi}{2}$-pulse along the -**y** direction rotates the spin back to the **z** axis for fluorescence detection (IV), and the measured photon count is designated as the bright signal, $S_B(t)$. By repeating the same

sequence but with a final $\frac{\pi}{2}$-pulse along the positive +**y** axis before readout, we measure the fluorescence of an orthogonal spin state to be the dark signal, $S_D(t)$. The difference between the two measurements, $C(t) = [S_B(t) - S_D(t)]/S_R(t)$, can faithfully represent the measured spin coherent dynamics of $V_B^-$, where $S_R(t)$ is a reference signal we measure at the end of the initialization laser pulse (II).

Figure 1e shows the measured spin echo dynamics of the highest dosage hBN sample S3. We find that the measured fluorescence contrast, $S_B(t)/S_R(t)$ [$S_D(t)/S_R(t)$], changes dramatically with different laser powers (inset), originating from the charge state relaxation dynamics after the laser pumping. This is particularly prominent at high laser power, where the optical ionization of the defect charge state is enhanced. This effect can lead to an artifact in the extracted spin echo timescales, which may explain the previous discrepancy in the measured $T_2^{Echo}$. However, the obtained fluorescence contrast from differential measurement, $C(t)$, is consistent across different laser powers, enabling an accurate extraction of the spin coherent timescales.

A few remarks are in order. First, this differential measurement scheme has been widely employed in the studies of the dense ensemble of NV centers in diamond to counter the ionization process[9,41,65–67]. Secondly, previous theoretical studies predict that the ionization of $V_B^-$ requires significantly higher energy ( ~ 4.46 eV) than the ionization of NV centers ( ~ 2.7 eV)[66,68,69]. This may explain why our experimental observation that the two-photon ionization process for $V_B^-$ only becomes evident under strong laser power ( ~ 10 mW); while the ionization of NV centers happens at ~ 10 − 20 $\mu$W laser[65,66]. Third, we note that unlike neutral $NV^0$ centers which emit fluorescence starting at 575 nm, neutral boron-vacancy $V_B^0$ has not been directly observed from photo-luminescence signals. Therefore the proposed ionization process only offers a potential explanation of the experiment.

### Coherent dynamics

Equipped with the robust differential measurement scheme, we now turn to the investigation of coherent dynamics of $V_B^-$ ensemble at

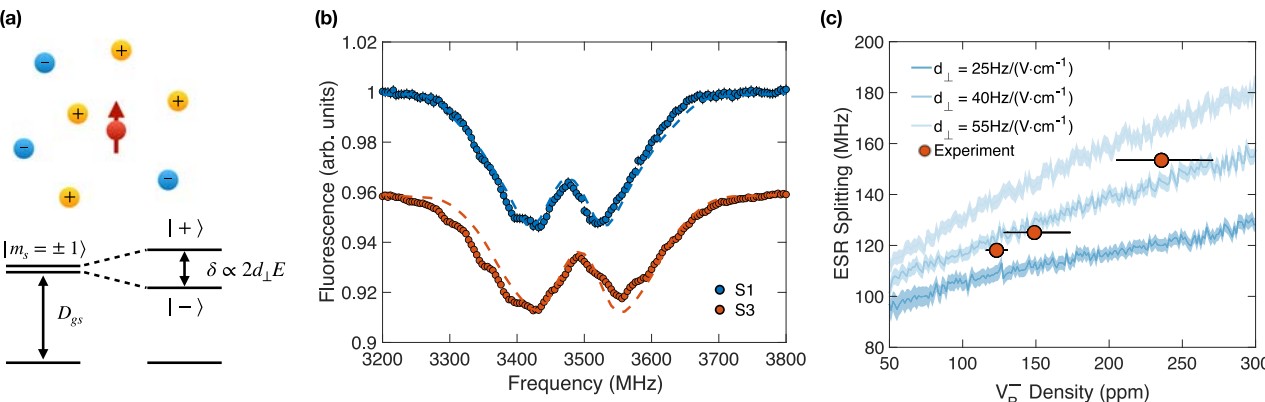

**Fig. 4 | Imaging the local electric field signals. a** Top: Schematic depicting the charged defects environment surrounding a $V_B^-$ electronic spin. Nearby negatively and positively charged defects create a local transverse electric field $E$ on $V_B^-$. Bottom: Energy level diagram of the $V_B^-$ spin state in the presence of the electric field: the E-field mixes the $|m_s = \pm 1\rangle$ states to new eigenstates $|\pm\rangle$, with a splitting, $\delta \propto 2d_\perp E$. **b** Measured ESR spectra of sample S1 and sample S3 at zero magnetic field. Dashed lines are the simulated results from our microscopic charged model using $d_\perp = 40$ Hz/(V cm$^{-1}$) and $V_B^-$ densities extracted from the previously measured coherent dynamics. Fluorescences are shifted vertically for comparison. **c** Numerically simulated ESR splitting $\delta$ using different electric susceptibilities, $d_\perp$. The red dots are the experimental results with error bars representing the range of densities whose residuals lie within 5% (see Methods).

various defect densities. The decoherence mechanism of $V_B^-$ consists of two major contributions: (1) the Ising coupling to the bath spins in the environment; (2) the dipolar interaction between $V_B^-$ ensemble themselves (Fig. 1a and Methods)[59]. To isolate the effect of each component, we measure the coherent dynamics of $V_B^-$ using three different dynamical decoupling pulse sequences.

We start with the spin echo pulse sequence, which is commonly used to characterize the coherent properties of a quantum system. Spin echo can decouple the static components of the Ising coupling between $V_B^-$ and the spin bath. By fitting the measured dynamics to a single exponential decay, $\sim e^{-(t/T_2^{\text{Echo}})}$, we extract $T_2^{\text{Echo}} \approx 70$ ns across all three hBN samples (Fig. 2b). This observation indicates that the spin echo decoherence of $V_B^-$ is predominantly limited by the spin fluctuation within the environmental spin bath, which does not depend on the $V_B^-$ concentration. Indeed, a previous study has shown that the Ising coupling to the local nuclear spin bath (nitrogen-14, boron-10, and boron-11), as well as the dark electronic spins, can accurately account for the measured spin echo timescales[52].

Next, we apply a more advanced dynamical decoupling pulse sequence, XY-8, to better decouple the $V_B^-$ ensemble from the environment. Instead of a single refocusing $\pi$-pulse, XY-8 employs a series of $\pi$-pulses with alternating phases (Fig. 1c). We fix the time intervals between pulses, $\tau_0 = 4$ ns, sufficiently smaller than the correlation timescale of the local spin bath (estimated from the spin echo timescale)[41,45]. As a result, XY-8 is expected to further suppress the fluctuations within the local spin noise and improve the measured spin coherent timescales. This is indeed borne out by our data. As shown in Fig. 2, the extracted coherence times, $T_2^{\text{XY8}}$, are significantly extended in all three samples. In contrast to the previous spin echo measurement where $T_2^{\text{Echo}}$ does not depend on $V_B^-$ density, here we observe that $T_2^{\text{XY8}} = [250 \pm 35]$ ns of sample S1 is longer than sample S3, $T_2^{\text{XY8}} = [167 \pm 10]$ ns. This suggests that $V_B^- - V_B^-$ interaction plays a key role in the measured XY-8 coherent timescales. Indeed, in XY-8 measurement, since the refocusing $\pi$-pulses flip all $V_B^-$ spins together, there is no suppression of the dipolar interaction between $V_B^-$ (see Methods).

To this end, we introduce DROID pulse sequence to further decouple the dipolar interaction within $V_B^-$ themselves (Fig. 1c)[59]. By applying a series of $\pi/2$ rotations along different spin axes to change the frames of interaction (also known as toggling frames), DROID modifies the dipolar Hamiltonian to an isotropic Heisenberg interaction, where the initial state, $\otimes_i \frac{|0\rangle_i + |-1\rangle_i}{\sqrt{2}}$, constitutes an eigenstate of the Heisenberg interaction, and consequently does not dephase (see Methods). As shown in Fig. 2, the measured coherent timescales, $T_2^D$,

indeed exhibit an approximate two-fold increase compared to $T_2^{\text{XY8}}$ across all three samples, agreeing with the cancellation of dipolar-induced decoherence.

Interestingly, we also observe that the spin relaxation time, $T_1$, and spin-locking time, $T_1^\rho$, both decrease with increasing ion implantation dosages (Fig. 2b). In principle, the dipolar interaction between $V_B^-$ will not lead to a decrease of $T_1$ due to the conservation of total spin polarization during the flip-flop process (see Supplementary Note 2.2). This $T_1$ related trend may be attributed to the presence of lattice damage during the implantation process or local charge state fluctuations[65]. We note that the spin relaxation process will introduce an additional decay to the coherent dynamics. However, the measured $T_1$ and $T_1^\rho$ are much longer than $T_2$ across all three samples at room temperature (Fig. 2). Nevertheless, we fix the duration between the polarization (II) and the read-out (IV) laser pulses to account for the effect of $T_1$ relaxation on the $T_2$ measurement (see Methods).

### Extracting $V_B^-$ density

The difference between $T_2^{\text{XY8}}$ and $T_2^D$ originates from the $V_B^- - V_B^-$ dipolar interaction, which can be used to estimate the density of $V_B^-$ directly. In particular, by randomly positioning 12 electronic spins at different sampling concentrations, we construct the dipolar interacting Hamiltonian of the system,

$$\mathcal{H}_{\text{dip}} = \sum_{i<j} -\frac{J_0 \mathcal{A}_{i,j}}{r_{i,j}^3}\left(S_i^z S_j^z - S_i^x S_j^x - S_i^y S_j^y\right), \quad (1)$$

where $J_0 = 52$ MHz·nm$^3$, $\mathcal{A}_{i,j}$ and $r_{i,j}$ represent the angular dependence and the distance between the $i^{th}$ and $j^{th}V_B^-$ spins, and $\{S_i^x, S_i^y, S_i^z\}$ are the spin-1/2 operators for $i^{th}V_B^-$ center (see Methods). By evolving the many-body system under different pulse sequences and averaging the spin coherent signals across random spin positional configurations, we obtain the simulated results of the corresponding XY-8 and DROID coherent timescales (Fig. 3a, see Methods)[9,65,70]. We observe from our simulation that both $T_2^{\text{XY8}}$ and $T_2^D$ indeed decrease with increasing $V_B^-$ density, while $T_2^D$ exhibits a longer timescale than $T_2^{\text{XY8}}$ across the density range surveyed. By minimizing the relative squared residuals of $T_2^{\text{XY8}}$ and $T_2^D$ between simulation and experiment, we estimate the $V_B^-$ concentration to be $\rho_{V_B^-}^{S1} \approx 123 \, ^{+8}_{-8}$ ppm, $\rho_{V_B^-}^{S2} \approx 149 \, ^{+25}_{-21}$ ppm, and $\rho_{V_B^-}^{S3} \approx 236 \, ^{+35}_{-31}$ ppm. The discrepancy between the measured

and simulated timescales may stem from imperfect spin rotations in the experiment, as well as finite-size effects from the simulations (see Methods). To further validate our $V_B^-$ density estimation, we measure the fluorescence count rates for the three hBN samples and find them to be proportional to the estimated $V_B^-$ densities $\rho_{V_B^-}$ (Fig. 3a inset, and Supplementary Table S1).

We highlight that although the ion implantation dosage spans nearly two orders of magnitude across three hBN samples, the estimated $V_B^-$ density only differs approximately by a factor of 2. This indicates that with larger implantation dosage, one may create more $V_B$ defects, but most of them remain charge neutral[61,62,66,71]. Using SRIM (Stopping and Range of Ions in Matter) program, we estimate the created $V_B$ defect density in the experiment to be $\rho_{V_B}^{S1} \approx (5.4 \pm 0.5) \times 10^2$ ppm, $\rho_{V_B}^{S2} \approx (2.0 \pm 0.2) \times 10^3$ ppm and $\rho_{V_B}^{S3} \approx (1.8 \pm 0.2) \times 10^4$ ppm, increasing linearly with the implantation dosage (see Methods). Figure 3b shows the negatively charged $V_B^-$ ratio, $\eta \equiv \rho_{V_B^-}/\rho_{V_B}$, which exhibits a substantial drop with increasing implantation dosages. This suggests that one may need to seek alternative solutions other than simply cranking up the irradiation dosage to achieve higher $V_B^-$ concentration for future applications in quantum information. We note that if one directly uses $\rho_{V_B}$ from SRIM to represent the negatively charged $V_B^-$ density, the simulated coherent timescales $T_2^{XY8}$ and $T_2^D$ will be significantly shorter than the experimental results (see Supplementary Fig. 3b).

## Probing the local charged defect environment

The presence of negatively charged $V_B^-$ ensemble in hBN also leads to a local electric field signal that can be directly probed using the spin degree of freedom of $V_B^-$ (Fig. 4a). Given the mirror symmetry of $V_B^-$ lattice structure respect to the $\mathbf{x} - \mathbf{y}$ plane, its electric field susceptibility along $\mathbf{z}$ vanishes, and one only needs to consider the transverse component of the local electric field. Without any external magnetic field, a transverse electric field to the $\mathbf{z}$-axis of $V_B^-$ (c-axis of hBN), $E_\perp$, will mix the original $|m_s = \pm 1\rangle$ states of $V_B^-$, and split them into two new eigenstates, $|\pm\rangle$[61,62,72,73]. To the leading order, the energy splitting, $\delta$, between $|\pm\rangle$ is proportional to the strength of the transverse electric field, $\delta \propto 2d_\perp E_\perp$, where $d_\perp$ is the ground state transverse electric field susceptibility of $V_B^-$ (Fig. 4a). In reality, the presence of the three first-shell $^{14}N$ nuclear spins as well as the intrinsic broadening of the $V_B^-$ transitions will lead to additional modification to the measured energy splitting $\delta$, and a detailed discussion of such effect can be found in Methods.

The splitting $\delta$ can be probed via the electron spin resonance (ESR) measurement: by sweeping the microwave field frequency and monitoring the fluorescence signals of $V_B^-$, one observes a fluorescence drop when the microwave is resonant with one of the spin transitions. Figure 4b shows the measured ESR spectra for sample S1 and S3 at zero magnetic field. Crucially, we observe that the splitting increases with $V_B^-$ concentration, consistent with the expectation that a higher charged defect density can generate a stronger local electric field. We also notice a small shift of the ESR center frequencies with increasing implantation dosages, which may originate from the implantation-induced strain effect[25,74–76].

To quantitatively understand the density dependence of the measured splitting, we utilize a microscopic model based upon randomly positioned electrical charges inside the hBN lattice. Such model has been successfully applied to capture the measured energy splitting between $|m_s = \pm 1\rangle$ sublevels of NV centers in diamond before[61,62]. Specifically, we randomly position charged defects surrounding a $V_B^-$ center at a density $\rho_c$, and calculate the corresponding transverse electric field $E_\perp$ at the $V_B^-$ site. Here we assume that these charges consist primarily of the negatively charged $V_B^-$ centers themselves (which are electron acceptors) and their associated donors − as a result, the local charged defect

density $\rho_c \approx 2\rho_{V_B^-}$. By diagonalizing the lab frame spin Hamiltonian in the absence of an external magnetic field (see Methods), we calculate the transition frequencies of the ESR experiment. The final simulated ESR spectrum is obtained via averaging over different charge defect configurations, as well as the spin states of the three closest hyperfine-coupled $^{14}N$ nuclear spins. Since $d_\perp$ of $V_B^-$ has not been determined before, we survey a range of different $d_\perp$ in our numerics to obtain a series of simulated ESR splitting at a variety of $V_B^-$ density (Fig. 4c). Comparing the experimentally measured ESR splitting $\delta$ to the simulated results from our model, we are able to get a rough estimation of the $V_B^-$ ground state transverse electric field susceptibility, $d_\perp \approx 40\,\mathrm{Hz}/(V \cdot cm^{-1})$. We note that the estimated $d_\perp$ of $V_B^-$ is on the same order of NV center in diamond, $d_\perp^{NV} \approx 17\,\mathrm{Hz}/(V \cdot cm^{-1})$[77].

## Outlook

Looking forward, our work opens the door to a number of intriguing directions. First, the characterization and control of coherent dipolar interaction in dense ensembles of spin defects in 2D materials represent the first step to using such platforms for exploring exotic many-body quantum dynamics. One particularly interesting example is to investigate the stability of phenomena such as many-body localization and Floquet thermalization in two and three dimensions. In fact, in long-range interacting systems, the precise criteria for delocalization remain an open question; whereas in Floquet systems, the thermalization dynamics involve a complex interplay between interaction and dimensionality[44,46,49,51]. Secondly, the measured low negatively charged $V_B^-$ ratio at high ion implantation dosage suggests that one may be able to use external electric gating to substantially tune and enhance the portion of $V_B^-$ concentration. Indeed, electric gating has been recently demonstrated as a powerful tool to engineer the charge state of optical spin defects in solid-state materials[78–81]. Finally, the estimated transverse electric field susceptibility highlights the potential use of $V_B^-$ as an embedded electric field sensor for in-situ characterization of heterogeneous materials[62,72,82,83].

## Methods
### hBN device fabrication
The sample consists of ion-irradiated hBN flakes and a titanium/gold (10/300 nm thick) coplanar waveguide (CPW) with a 50 μm wide central stripline, on a sapphire substrate. hBN flakes were tap-exfoliated from a commercially available hBN single crystal and transferred onto Si substrates. Boron vacancy defects were generated using He$^+$ ion implantation with an energy of 3 keV with dose densities, $0.30 \pm 0.03$ nm$^{-2}$ (sample S1), $1.1 \pm 0.1$ nm$^{-2}$ (sample S2), and $10 \pm 1$ nm$^{-2}$ (sample S3), respectively. Error bars on the dosages account for measured current fluctuations during the implantation process. After ion irradiation, hBN flakes were transferred on top of the CPW using the PC/PDMS transfer method[31,53].

### Sources of decoherence
The sources of $V_B^-$ decoherence have two major contributions: (1) the Ising coupling to the environmental spin bath, such as nuclear spins and dark electronic spins, and (2) the dipolar interaction within the $V_B^-$ ensemble[52]. In particular, the interaction between a $V_B^-$ center and the local off-resonant spin bath takes the form of Ising coupling (under rotating-wave approximation)

$$\mathcal{H}_{Ising} = \sum_k A_k^{zz} S^z \mathfrak{S}_k^z = \left(\sum_k A_k^{zz} \mathfrak{S}_k^z\right) S^z, \qquad (2)$$

where $A_k^{zz}$ represents the strength of the Ising coupling between $V_B^-$ and the $k$th bath spin, $\mathfrak{S}_k^z$ is the spin operator for the $k$th bath spin, and $S^z$ is the spin-1/2 operator for $V_B^-$ when restricting to the spin subspace $|m_s = 0\rangle, |m_s = -1\rangle$ (See Supplementary Note 2.1).

A few remarks are in order. First, summing over the Ising coupling to bath spins results in a random on-site field disorder on each $V_B^-$, $h = \sum_k -\frac{J_0 \mathcal{A}_k}{r_k^3} \mathfrak{S}_k^z$, which would effectively broaden the electronic spin transition. Second, the strength of the Ising coupling consists of two different contributions: Fermi contact interaction and dipolar interaction. For the closest three spin-1 $^{14}N$ nuclear spins, $A^{zz} \approx 47$ MHz is dominated by Fermi contact, which leads to the previously reported seven-peaks hyperfine structure in the measured ESR spectrum[31]. On top of that, the interaction with other far away bath spins like $^{10}B$, $^{11}B$ and $^{14}N$ nuclear spins, as well as dark electronic spins[52], would further broaden the ESR spectrum. Indeed in our experiment, the measured $V_B^-$ ESR peak associated with the spin transition $|m_s = 0\rangle \Longleftrightarrow |m_s = -1\rangle$ (under an external magnetic field ~ 250 G) exhibits a large linewidth which can be well captured using a Gaussian function. The extracted standard deviation of the ESR resonance is around 80 MHz across all three hBN samples studied in this work (Supplementary Fig. 2a), independent of the ion implantation dosages. Last, given the large broadening of the transition, one requires a strong microwave field to efficiently drive the spin state of $V_B^-$. In our experiment, we utilize microwave pulses with a Rabi frequency $\Omega \approx 83$ MHz ($\pi$ − pulse duration $t_\pi = 6$ ns), similar to the measured $V_B^-$ linewidth, to achieve rapid spin manipulation with reasonable fidelity (Supplementary Fig. 2b).

The other source for decoherence is the dipolar interaction within the $V_B^-$ spin ensemble. In the rotating frame, the Hamiltonian that governs the dipolar interaction of $V_B^-$ can be written as

$$\mathcal{H}_{dip} = \sum_{i<j} -\frac{J_0 \mathcal{A}_{i,j}}{r_{i,j}^3} \left( S_i^z S_j^z - S_i^x S_j^x - S_i^y S_j^y \right), \qquad (3)$$

where $J_0 = 52$ MHz nm³, $\mathcal{A}_{i,j}$ and $r_{i,j}$ represent the angular dependence and the distance between the $i^{th}$ and $j^{th}$ $V_B^-$ spins, and $S_i^x, S_i^y, S_i^z$ are the spin-1/2 operators for $i^{th}$ $V_B^-$ centers. We note that $\mathcal{H}_{dip}$ corresponds to the energy-conserving terms of the dipolar interaction, i.e., the rotating-wave approximation, when restricting our attention to the $V_B^-$ spin subspace $|m_s = 0\rangle$, $|m_s = -1\rangle$ (See Supplementary Note 2.1). For the highest $V_B^-$ density sample S3 in this work, $\rho_{S3} \approx 236^{+35}_{-31}$ ppm, we estimate the average dipolar interaction strength between nearby spins to be $\langle J \rangle$ ~ 1.2 MHz.

### Dynamical decoupling sequences

To selectively isolate the effect of each decoherence source, we introduce three different dynamical decoupling sequences to investigate the coherent dynamics of $V_B^-$ ensemble. We start with the most basic sequence, Spin Echo, which applies a single refocusing $\pi$-pulse at the center of the time evolution (Supplementary Fig. 3b). This refocusing pulse reverses the on-site field disorder (Ising coupling in Eq. S1) for the second half of the time evolution, $\sum_i h_i S_i^z \longrightarrow \sum_i -h_i S_i^z$, thus negating the accumulated phase from the static component of the on-site disorder. However, Spin Echo cannot decouple the decoherence that arises from the time-dependent fluctuation of the on-site random field. For the case of the $V_B^-$, Spin Echo decay originates from the internal spin flip-flops within the environmental spin bath. As a result, the Spin Echo timescale $T_2^{Echo} \approx 70$ ns offers a good estimation of the correlation time, $\tau_c$, of the spin bath.

Next, the more advanced decoupling sequence XY-8 also employs refocusing $\pi$-pulse, but, instead of one, XY-8 applies a series of $\pi$-pulses to further isolate the $V_B^-$ from the environment (Supplementary Fig. 3c). The alternating phase of $\pi$-pulses along different spin axes (X-Y-X-Y-Y-X-Y-X) is designed to suppress intrinsic pulse errors to higher order. We fix the interval time between $\pi$-pulses to be 4 ns, much shorter than the correlation time $\tau_c$ of the spin bath, and sweep the total pulse number. As expected, using XY-8 pulse sequence, the local bath spin fluctuation is further decoupled from the $V_B^-$ ensemble and we observe significant extensions of the measured coherence times across all three hBN samples.

However, the decoupling mechanics of XY-8 does not apply to the dipolar interaction within the $V_B^-$ ensemble, $\mathcal{H}_{dip}$. The intuition is simple. The microwave $\pi$-pulses flip all the $V_B^-$ spins together, and the two-body terms in $\mathcal{H}_{dip}$ remains unchanged. For instance, under a rotation along the **x** axis, $\mathcal{H}_{dip}$ becomes $\mathcal{H}'_{dip} = \sum_{i<j} -\frac{J_0 \mathcal{A}_{i,j}}{r_{i,j}^3} [(-S_i^z)(-S_j^z) - S_i^x S_j^x - (-S_i^y)(-S_j^y)] = \mathcal{H}_{dip}$.

This is where the specifically designed interaction decoupling sequence, DROID (Disorder-RObust Interaction-Decoupling), comes in[59]. In particular, by applying a series of $\pi/2$ rotations along different spin axes to periodically change the frames of interaction (also known as the toggling frame), the leading-order effective Hamiltonian can be described by a simple weighted average of each toggling-frame Hamiltonian (Supplementary Fig. 3d). For spin-1 $V_B^-$, the resulting effective Hamiltonian, $\mathcal{H}_{eff}$, takes the form of an isotropic long-range Heisenberg interaction,

$$\mathcal{H}_{eff} = \sum_{i<j} \frac{J_0 \mathcal{A}_{i,j}}{r_{i,j}^3} \frac{1}{3} \left( S_i^z S_j^z + S_i^x S_j^x + S_i^y S_j^y \right) = \sum_{i<j} \frac{J_0 \mathcal{A}_{i,j}}{r_{i,j}^3} \frac{1}{3} \hat{\mathbf{S}}_i \cdot \hat{\mathbf{S}}_j \qquad (4)$$

Since our initial spin state, $\otimes_i \frac{|0\rangle_i + |-1\rangle_i}{\sqrt{2}}$, constitutes an eigenstate of the above Heisenberg Hamiltonian, DROID sequence further suppresses the decoherence effect originating from the dipolar interaction within $V_B^-$ ensemble. This is indeed borne out by our data. The measured coherent timescales under DROID sequence exhibit an additional 2-fold increment compared with XY-8 sequence across all three hBN samples.

### Numerical simulation of the coherent dynamics

To quantitatively analyze the coherent timescales measured from the three different samples, we perform numerical simulation using 12 quantum spins randomly positioned onto the hBN lattice with varying density $\rho$. For the dipolar interaction between $V_B^-$, we build up the Hamiltonian for each pair of spins using the form in Eqn. (3). As for the on-site random field disorder, we introduce an additional term $\sum_i h_i S_i^z$ into the Hamiltonian, with $h_i$ drawn from a Gaussian distribution with standard deviation 80 MHz which is independently characterized from the $V_B^-$ ESR resonance spectrum (Supplementary Fig. 2a). To account for the effect of finite pulse duration in the experimental sequence, a microwave driving term $\pm \Omega \sum_i S_i^{x(y)}$ is included into the simulation whenever a pulse with specific phase is applied, where $\Omega = 83$ MHz is directly determined using the corresponding Rabi oscillation measurement (Supplementary Fig. 2b). For each pulse sequence, respective $\pi$- and $\frac{\pi}{2}$-pulses and intervals are applied as in the experimental procedure, and we also utilize the differential measurement scheme in the simulation to faithfully capture the experimental details. After time evolution, we only use the central spin's polarization to represent the coherent dynamics, as the far away spins would suffer from significant finite-size effects due to the small system size that one can compute in numerics. At a given $V_B^-$ density $\rho_{V_B^-}$, we average for 1000 disorder realizations of random spin positions and on-site fields to obtain the final simulated curve.

To efficiently compute the corresponding quantum dynamics, we employ DYNAMITE, a powerful package providing a simple interface to fast evolution of quantum dynamics[84]. In contrast to the traditional Hamiltonian diagonalization method that requires exponentially increasing time as spin number goes up, DYNAMITE uses the PETSc/SLEPc implementations of Krylov subspace exponentiation and eigensolving, which drastically lower the computational resources for simulations involving large spin number.

## $V_B^-$ Density extraction

**Calculation of residuals.** In order to extract the $V_B^-$ density, we compare the experimentally measured XY-8 and DROID coherent timescales to the numerical simulations. In particular, we first fit the coherence time $\mathcal{T}_2$ to $\mathcal{T}_2 \propto \rho^{-\alpha}$, where $\rho$ is the $V_B^-$ density, and $m$ is a free fitting parameter. This fitted line should be a straight line in our log-log plot (Supplementary Fig. 4a), where the slope corresponds to the value of $\alpha$. We find the fitted value of $\alpha \sim 0.8 \pm 0.1$ for XY-8 to be fairly close to one, and this validates that the main decoherence source for XY8 is from the dipolar interaction whose strength scales linearly with $V_B^-$ density. Note that $\mathcal{T}_2^D$ theoretically should be unchanged with increasing $V_B^-$ density since the DROID sequence should decouple the $V_B^-$ dipolar interaction, but due to other factors like imperfect drive the decoherence timescale still decrease. To extract the $V_B^-$ densities, we calculate the sum of squared relative residuals of XY-8 and DROID between the experimental values and the fitted $\mathcal{T}$.

$$\text{Residuals} = \left(\frac{\log \mathcal{T}_2^{XY8} - \log T_2^{XY8}}{\log T_2^{XY8}}\right)^2 + \left(\frac{\log \mathcal{T}_2^D - \log T_2^D}{\log T_2^D}\right)^2 \quad (5)$$

Here, we adopt the minimums of the residual curves as our estimated density and estimate their errors from the range values whose residuals lie within 5% of the minimum (Supplementary Fig. 4b blue shaded regions). This put the estimated densities for the three samples to be $123^{+8}_{-8}$, $149^{+25}_{-21}$, and $236^{+35}_{-31}$ ppm respectively. We note the discrepancy between the measured and simulated timescales may stem from (1) pulse imperfections in the experiment, and (2) finite-size effect, i.e., how many spins one can simulate, in numerical simulations. The presence of nearby bath spins (nuclear spins and dark electronic spins) leads to a large broadening of the $V_B^-$ transition with measured resonance linewidth $\sigma \sim 80$ MHz (standard deviation). Therefore, with Rabi driving frequency $\Omega \approx 83$ MHz, which is comparable to the broadening $\sigma$, the applied pulses will have significant imperfections. To better capture this effect in simulation, we incorporate a random static on-site field on each spin drawn from the measured resonance distribution, and average across all different configurations. However, the static on-site field does not portray the whole picture of the real experiment, where the bath spins also have dynamics. Due to the limited system size we can compute, it is particularly difficult to simulation this effect.

The measured $T_2$ decay in the experiment is a combination of both decoherence and imperfect pulse effect. Therefore, for DROID sequence where one decouples a large portion of the interactions, such pulse imperfection becomes more evident compared to XY8 sequence where the $T_2$ decay is governed by the dipolar interactions within $V_B^-$. This also explains why the discrepancy between experiment and simulation is larger in DROID $T_2$ compared to XY-8 $T_2$.

**Fluorescence counts.** To verify our estimation of the $V_B^-$ densities, we also measure the fluorescence counts $N_{tot}$ of the three samples with excitation laser of the same power ($\sim 0.4$ mW). The resulting counts are $(0.64 \pm 0.03) \times 10^6$, $(0.84 \pm 0.03) \times 10^6$, and $(1.44 \pm 0.04) \times 10^6$ photons/s respectively for sample S1, S2, and S3 (see Supplementary Table 1). It is important to note that this total counts $N_{tot}$ also include the noise background.

To account for this non-negligible background, we define the adjusted counts by

$$N_A = N_{tot} \times C, \quad (6)$$

where $C$ is the measured ESR contrast for the three samples. Here we also normalize $N_A$ to 1 for sample S1 for better comparison, and the

resulting adjusted counts across three samples do closely follow the $V_B^-$ densities estimated from spin coherent dynamics (see Main Text Fig. 3a inset).

## SRIM simulation

To estimate the total created boron-vacancy $V_B$ defect density at different ion implantation dosages, we perform a detailed calculation with full damage cascades using Stopping and Range of Ions in Matter (SRIM)[63]. In particular, we choose the incident beam to be Helium ion with the energy of 3 keV, targeting hBN layer with a thickness 100 nm. The simulated damage events distribution is shown in Supplementary Fig. 5a. We find that the created vacancies are distributed within the initial $\sim 60$ nm of the hBN sample. By integrating target vacancies at different depths, we conclude that each ion on average creates $\sim 11$ boron vacancies. For sample S1, S2, S3 with ion implantation dosages of $0.30 \pm 0.03$ nm$^{-2}$, $1.1 \pm 0.1$ nm$^{-2}$, and $10 \pm 1$ nm$^{-2}$, the created area density of boron vacancy is around $3.3 \pm 0.3$, $12.1 \pm 1.2$, and $110 \pm 10$ nm$^{-2}$ respectively. Using the atomic number density for hBN, 101.9 atom nm$^{-3}$, we estimate the total created boron vacancy density in sample S1, S2 and S3 to be $\sim (5.4 \pm 0.5) \times 10^2$, $(2.0 \pm 0.2) \times 10^3$ and $(1.8 \pm 0.2) \times 10^4$ ppm.

In main text Fig. 3, by comparing the experimentally measured coherent timescales to our numerical simulations, we conclude that only a small portion of the created boron-vacancy centers ($V_B$) from ion implantation process are in the desired negatively charged state ($V_B^-$). If one naively uses the $V_B$ density estimated from SRIM to represent $V_B^-$ concentration, we find that the simulated timescales are much shorter than the experimental results for all three hBN samples investigated in this work (Supplementary Figure 5), indicating significant overestimation of the $V_B^-$ concentration. Note that when the electronic spin density exceeds $\sim 1000$ ppm in our simulation, due to the strong dipolar interaction within the spin system, our finite-duration microwave pulses cannot faithfully drive the spin anymore. As a result, the simulation exhibits nearly vanishing spin polarization even after the first set of driving pulses. Here we use a dashed line to extend the DROID simulation above $\sim 1000$ ppm for better comparison.

## Numerical simulation of ESR splittings and extraction of transverse electric field susceptibility

In this section, motivated by pioneer studies of nitrogen-vacancy centers in diamond[61,62,73], we model the measured $V_B^-$ ESR splitting in the absence of any external magnetic field using a microscopic model based upon the local charged defects in hBN. In particular, the presence of randomly distributed charges will create a vector electric field $\vec{E}$ at the site of each $V_B^-$. Intuitively, we expect these charges to consist primarily of the negatively charged $V_B^-$ centers themselves (which are electron acceptors) and their associated donors — as a result, the local charged defect density $\rho_c \approx 2\rho_{V_B^-}$ with $\rho_{V_B^-}$ the $V_B^-$ defect density. The corresponding $V_B^-$ electronic spin-1 ground state Hamiltonian can be written as:

$$\mathcal{H} = D_{gs}S_z^2 + \Pi_x\left(S_y^2 - S_x^2\right) + \Pi_y(S_xS_y + S_yS_x) + \sum_{i=1}^{3} A_{zz}I_z^iS_z, \quad (7)$$

where $D_{gs} \approx 3.48$ GHz is the zero-field splitting between $|m_s = 0\rangle$ and $|m_s = \pm 1\rangle$ spin levels, $\Pi_{\{x,y\}} = d_\perp E_{\{x,y\}}$ characterizes the $V_B^-$ coupling to local perpendicular electric field with susceptibility $d_\perp$, $S$ and $I^i$ are the spin-1 operators for $V_B^-$ electronic spin and the closest three $^{14}$N nuclear spins respectively, and $A_{zz} \approx 47$ MHz is the hyperfine coupling strength[31,52].

A few remarks are in order. First, the **z** is defined along the c-axis of hBN (perpendicular to the lattice plane), **x** and **y** lie in the lattice plane, with **x** oriented along one of the three vacancy-nitrogen bonds. Due to the mirror symmetry of $V_B^-$ with respect to the **x** − **y** plane, its electric

field susceptibility in **z** vanishes, $d_\parallel = 0$[85]. Second, the presence of $^{14}$N nuclear spins generate an effective local magnetic field along **z** direction on the $V_B^-$, whose strength depends on the specific nuclear spin states. Since the three nuclear spins are all in fully mixed states, we can simply set each $I_z^i = \{-1, 0, +1\}$ with equal probabilities, and average across different nuclear spin configurations to account for their contributions.

To quantitatively capture the experimentally measured ESR spectrum at zero magnetic field, we randomly sample the charged defects in space at a given density $\rho_c \approx 2\rho_{V_B^-}$ and calculate the perpendicular electric field, $E_{\{x,y\}}$, at the $V_B^-$ site from the closest 10 sampled charges. By diagonalizing the Hamiltonian in Eqn. (7), we obtain the corresponding eigenstates and the associated eigenenergies, with which we can back out the resonances in ESR. To account for the intrinsic broadening of the spin resonance, we convolve each resonance with a Gaussian profile with standard deviation $\sigma \approx 25$ MHz to better capture the experimental data. The final simulated ESR spectra are obtained by averaging over 1000 different randomly positioned charge configurations.

Supplementary Fig. 6a shows the simulated ESR spectra on top of the experimental data using $\rho_{V_B^-} = \{123^{+8}_{-8}, 149^{+25}_{-21}, 236^{+35}_{-31}\}$ ppm for sample S1, S2, and S3, and $d_\perp = 40$ Hz/(V·cm$^{-1}$). The excellent agreement between the numerical simulation and experiment highlights the validity of our microscopic charge model. We remark that, given the large intrinsic broadening of the spin transitions, the measured ESR spectra can be also well captured with a sum of two Lorentzian profiles, as evinced in Supplementary Fig. 6b, with which we extract the values of the ESR splitting $\delta$ in the main text Fig. 4c.

## Data availability
Source data are included with this published article. Further data are available from the corresponding author upon reasonable request. Source data are provided with this paper.

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

## Acknowledgements

We gratefully acknowledge the insights of and discussions with N.Y. Yao, C. Dai, J. Kruppe, P. Zhou, E. Davis, B. Kobrin, V. Liu, W. Wu, K. W. Murch, L. Yang, D. Li, and H. Zhou. We thank G. Kahanamoku-Meyer and S. Iyer for their assistance in setting up numerical simulations. This work is supported by the Startup Fund, the Center for Quantum Leaps, the Institute of Materials Science and Engineering, and the OVCR Seed Grant from Washington University. E. A. Henriksen acknowledges support from NSF CAREER DMR-1945278 and AFOSR/ONR DEPSCOR no. FA9550-22-1-0340. T. Li acknowledges support from the DARPA ARRIVE program and the NSF under grant no. PHY-2110591.

## Author contributions

C.Z. conceived the idea. R.G., G.H., and Z.L. performed the experiment and analyzed the data. R.G., B.Y., and C.Z. developed the theoretical models and performed the numerical simulations. X.G., P.J., E.A.H., and T.L. fabricated the hBN samples. R.G. and C.Z. wrote the manuscript with inputs from all authors.

## Competing interests

The authors declare no competing interests.
