## [Peer Review File · Nature Communications]

REVIEWER COMMENTS

Reviewer #1 (Remarks to the Author):

This manuscript, entitled “Coherent Dynamics of Strongly Interacting Electronic Spin Defects in Hexagonal Boron Nitride” by R. Gong et al., reports experimental results in which dynamical decoupling sequences are applied to ensembles of optically active negatively-charged boron-vacancy (VB-) centers in order to determine their ground state spin coherence times, the primary sources of decoherence, and the defect density in three samples implanted with different dosages. The primary new results are the use of several different pulse sequences and a differential measurement technique (all of which have been used before with NV centers in diamond) to determine the role of dipolar interactions between VB- centers in decoherence, and the use of electron spin resonance at zero magnetic field to study the charge environment of the VB- centers and to estimate their transverse electric field susceptibility.

In my opinion this manuscript is well-written (aside from minor grammatical errors and typos here and there that should be corrected in copy-editing), the results are for the most part clearly presented, and the methods and experimental data all appear to be sound. These results are of considerable interest to experts working with defects in 2D materials, and more broadly will be of some interest to anyone interested in solid-state quantum system. I feel these results could be made appropriate for publication in Nature Communications.

However, I am not entirely convinced by some of the conclusions presented in the manuscript, and I have a number of questions and comments (detailed below) that I feel the authors should address. I therefore recommend the authors revise their manuscript to address these questions, and either strengthen their justifications or soften their claims.

Specific comments and questions:

- 1) My biggest concern is with regards to the usage of the XY-8 and DROID sequences to “directly determine the precise concentration of VB-,” to quote from the abstract. This is presented as a central result of the paper, and to be frank, it doesn’t feel entirely justified as the agreement between the theory and experiment shown in Figure 3 does not seem that good. In addition, I don’t really understand the justification for using the geometric means of the T2 values for XY-8 and DROID to match the simulations and experiment. While I agree that the observed trends are

suggestive and that there seems to be some degree of qualitative agreement, it seems fairly clear that the model used here is incomplete (see comment 3 below) and does not provide quantitative agreement, so I therefore think it is too strong a claim to say that the authors can precisely or quantitatively determine the VB- concentration based on this data.

2) Related to point 1, I feel that the data points in Figure 3B should have error bars indicating the uncertainty in both the x and y directions (along x from the uncertainty in SRIM, and uncertainty in y from both the experimental error bars on the T2 times and the disagreement with the model, which could presumably be accounted for using a chi-squared test or something similar and inflating them accordingly. I feel it would also be appropriate to include these error bars when quoting the VB- densities throughout the main text of the manuscript (for example, lines 171-172 of this draft.)

3) Given the model presented by the authors, I find it confusing that the XY-8 T2 times appear to be roughly the same for S1 and S2, while the DROID T2 times appear to be about the same for S2 and S3 (at least within error bars based on Figure 1e and Figure 3a). To me this seems to at least partially invalidate the conclusion that VB- dipolar coupling is primarily responsible for decoherence under the XY-8 sequence, given the different implantation dosages and different extracted VB- densities across all the 3 samples. Do the authors have an explanation for this?

4) The data set that seems to show the clearest trend with implantation dosage is actually T1, but the authors don't seem to address this at all. Is this consistent with their model?

5) I would think the authors could use a combination of the measured fluorescence rates and the contrast differences between SB, SD, and SR to independently extract the VB- center densities in each sample and either validate or invalidate their extracted VB- densities. Is this analysis possible? If so, I feel it should be done. If not, why not?

6) When I first saw Figure 1a, I initially believed that experiments had been performed on VB- centers in monolayers of hBN, but after carefully reading the manuscript I believe this is not the case. I think Figure 1a should be modified to make it clear that these experiments are carried out in 100 nm thick hBN flakes with many layers, and that the dipolar interactions (and the spin bath) could include coupling to electronic and nuclear spins in other layers.

7) On line 128 the authors write "We fix the time intervals between pulses, $\tau_0 = 4$ ns, sufficiently smaller than the correlation timescale of the local spin bath (estimated from the spin echo timescale)..." How then is the XY-8 time varied in Figure 1d? Does this mean the number of pulses is increased for each subsequent XY-8 data point along the x axis, rather than increasing the free

precession time between pi pulses? If so, I believe this is a fairly non-standard way of extracting T2 with dynamical decoupling sequences like XY-8, and this should also be stated explicitly and more clearly explained.

8) The authors write: “We note that the estimated d_{\perp} of V_B^- is on the same order of NV center in diamond...” Is this to be expected? Do the transverse electric field susceptibilities have similar origins in these two defects?

Reviewer #2 (Remarks to the Author):

The Authors report on experimental measurements of an ensemble of interaction negatively charged boron vacancies in hexagonal boron nitride, a prominent two-dimensional material being investigated as a host for quantum defects. The Authors demonstrate a few results. They show that the coherence times for the measured samples are dominated by two effects, the coupling to the nuclear bath and the inter-defect dipolar interactions. This is shown by using two different decoupling sequences targeting the different interactions. From numerical simulations, the authors can estimate the percentage of defects created in the negative charge state from implantation. I find the results to be interesting and worthy of publication. I just have two minor questions to be addressed.

1) “This is particularly prominent at high laser power, where the optical ionization of the defect charge state is enhanced.”

Calculations of the boron vacancy put the charge state transition levels far from the band edges (see e.g. <https://doi.org/10.1103/PhysRevB.97.214104>). Even a two-photon ionization process seems unlikely given the energetics. Are there any alternative processes that might explain the observations given the unlikely ionization?

2) It is unclear to me whether the statement that “< 5% of the created boron vacancy defects are in the desired negatively charged state” comes from simulations or experimental results. Given that there is some discrepancy between the numerical simulations and the exact experimental coherences that are measured, how reliable is the number? Perhaps the Authors can clarify this.

Reviewer #3 (Remarks to the Author):

In this paper the spin coherence properties of VB in hBN have been studied. Based on the decoherence mechanism of VB ensembles in these samples, the ratio of negatively charged VB to total boron vacancy defects has been modelled. At the end the electric field susceptibility of VB⁻ is calculated by comparing the splitting of E with difference density of VB. The decoherence mechanism has been reported in couple of works so far. The experiment has been done precisely and data have been analysed in details. The manuscripts could be organised better so the reader can follow the story. Currently the text jumps between figures which confuses the reader. Overall there are few important points that need clarification for this draft to be suitable for publication in a nature communication journal:

The authors referred to charge state and optical ionization in the first part of manuscript and to explain the results in figure 2. Is there any experimental evidence on the charge state switching of negatively charged Vb? Can the effects be explained considering initialisation rate to spin 0 rather than charge state switching?

Authors mentioned that the discrepancy between the measured and simulated timescales may stem from imperfect spin rotations in the experiment, as well as finite-size effects from the simulations. Can they comment on why the simulation does not match to the experimental data. Even the relative rates are different in the simulations. Both experimental data show similar drop in the coherence time vs VB concentration but this is not the case in simulation result. In fact, the TXY8 drops very fast in the simulation and there is only very small change in the TDRIOD vs VB concentration.

It is important to provide VB PL intensity vs implantation fluences and compare it to the result in this paper. Does it match with the proposed ratio of VB⁻ upon implantation?

From Figure 4 it is clear that there is also zero field splitting shift in the ODMR vs fluences. Is there any effect of strain on the results in this paper? I suggest to add this discussion in the manuscript.

We sincerely thank all the referees for their tremendously valuable comments, questions, and suggestions. We are especially grateful for all three referees being interested in our result and supportive of the publication, commenting that “(Referee 1) These results are of considerable interest to experts working with defects in 2D materials, and more broadly will be of some interest to anyone interested in solid-state quantum system”, “(Referee 2) I find the results to be interesting and worthy of publication”, and “(Referee 3) The experiment has been done precisely and data have been analysed in details”.

However, the referees also bring up a number of very important questions regarding the data analysis, the discrepancy between experiments and simulations, the proposed two-photon ionization process, the origin of the estimated electric field susceptibility, as well as the organization of the manuscript. We totally agree with the referees and apologize for not being clear and precise in the original manuscript. We have carefully re-analyzed the experimental results, as well as performing some additional measurements to justify our observation more clearly. Finally, we have tried our very best to carefully respond to all of the referee’s suggestions; this has led to some extensive rewriting of the manuscript as well as collections of new data. We truly believe that the manuscript we now resubmit is substantially improved thanks to all the referee’s comments, and we hope the referees will support publication in Nature Communications.

Reviewers Comments:

Reviewer #1 (Remarks to the Author):

This manuscript, entitled “Coherent Dynamics of Strongly Interacting Electronic Spin Defects in Hexagonal Boron Nitride” by R. Gong et al., reports experimental results in which dynamical decoupling sequences are applied to ensembles of optically active negatively-charged boron-vacancy (V_B^-) centers in order to determine their ground state spin coherence times, the primary sources of decoherence, and the defect density in three samples implanted with different dosages. The primary new results are the use of several different pulse sequences and a differential measurement technique (all of which have been used before with NV centers in diamond) to determine the role of dipolar interactions between V_B^- centers in

decoherence, and the use of electron spin resonance at zero magnetic field to study the charge environment of the V_B^- centers and to estimate their transverse electric field susceptibility.

In my opinion this manuscript is well-written (aside from minor grammatical errors and typos here and there that should be corrected in copy-editing), the results are for the most part clearly presented, and the methods and experimental data all appear to be sound. These results are of considerable interest to experts working with defects in 2D materials, and more broadly will be of some interest to anyone interested in solid-state quantum system. I feel these results could be made appropriate for publication in Nature Communications.

However, I am not entirely convinced by some of the conclusions presented in the manuscript, and I have a number of questions and comments (detailed below) that I feel the authors should address. I therefore recommend the authors revise their manuscript to address these questions, and either strengthen their justifications or soften their claims.

We must thank the referee for the thorough review of our manuscript and endorsement for publication. We are genuinely grateful for the invaluable comments and questions which have prompted us to better contextualize the results and polish our claims. Following these suggestions, we have carefully re-analyzed the experimental results, taken some additional data and made significant changes to the manuscript to further refine our experimental observations/conclusions.

Specific comments and questions:

1) My biggest concern is with regards to the usage of the XY-8 and DROID sequences to “directly determine the precise concentration of V_B^- ,” to quote from the abstract. This is presented as a central result of the paper, and to be frank, it doesn’t feel entirely justified as the agreement between the theory and experiment shown in Figure 3 does not seem that good. In addition, I don’t really understand the justification for using the geometric means of the T2 values for XY-8 and DROID to match the simulations and experiment. While I agree that the observed trends are suggestive and that there seems to be some degree of qualitative agreement, it seems fairly clear that the model used here is incomplete (see comment 3 below) and does not provide quantitative agreement, so I therefore think it is too strong a claim to say that the authors can precisely or quantitatively determine the V_B^- concentration based on this data.

We completely agree with the referee that our previous claim of precisely determining the V_B^- concentration is too strong. In all honesty, this is something we struggled with the most when writing the manuscript. To better capture our experimental result, we attempted various models and employed computer cluster to simulate a quantum system with $N = 12$ spins and many disorder averages, as well as including the finite pulse duration and the measured inhomogeneous broadening of V_B^- into our simulation, yet the simulation cannot completely reproduce the experiment results.

We believe the discrepancy mainly stems from (1) pulse imperfections in the experiment, and (2) finite-size effect, i.e. how many spins one can simulate, in numerics. The presence of nearby bath spins (nuclear spins and dark electronic spins) leads to a large broadening of the V_B^- transition with measured resonance linewidth $\sigma \sim 80$ MHz (standard deviation). Therefore, with our current Rabi driving frequency $\Omega \approx 83$ MHz in the experiment, which is comparable to the broadening σ , the applied pulses will have significant imperfections. To better capture this effect in simulation, we have already incorporated a random static on-site field on each spin drawn from the measured resonance distribution, and average across all different configurations to obtain the decay profiles from theory. However, the static on-site field does not portray the whole picture of the real experiment, where the bath spins also have dynamics. Due to the limited system size we can compute, it is extremely difficult for us to really capture such effect from numerical simulations.

The measured T_2 decay in the experiment is a combination of both decoherence and imperfect pulse effect. For DROID sequence where one decouples a large portion of the interactions, such pulse imperfection becomes more evident compared to XY8 sequence where the T_2 decay is governed by the dipolar interactions within V_B^- . This explains that the discrepancy between experiment and simulation is larger in DROID T_2 compared to XY-8 T_2 .

Therefore, we totally agree with the referee that our model has some limitations, and our previous claim is too strong. Correspondingly, we have added a short discussion in both the main text and methods to clarify the limitations of our model, as well as soften our claims in the abstract that “we directly estimate

the concentration of V_B^- .”

In addition, we agree with the Referee that using the geometric mean to match the experiment and simulation is not well justified, especially given the imperfections in our modeling. To this end, we have adopted a new density extraction procedure and incorporated the estimated error bars to account for the uncertainties in V_B^- density (see the reply below).

2) Related to point 1, I feel that the data points in Figure 3B should have error bars indicating the uncertainty in both the x and y directions (along x from the uncertainty in SRIM, and uncertainty in y from both the experimental error bars on the T2 times and the disagreement with the model, which could presumably be accounted for using a chi-squared test or something similar and inflating them accordingly. I feel it would also be appropriate to include these error bars when quoting the V_B^- densities throughout the main text of the manuscript (for example, lines 171-172 of this draft.)

We thank the referee for the super helpful comments, and apologize for not including the error bars in our original analysis. Following the referee’s suggestions, we re-analyze our measured and simulated timescales by minimizing the relative least-squared residuals to re-extract the estimated V_B^- density. In particular, we first fit our simulated coherent timescales T_2^{XY8} and T_2^D using a power-law function with V_B^- sampling density $T_2 \sim 1/\rho^\alpha$ (dashed lines in R2). For the XY8 sequence, the fitted $\alpha = 0.8 \pm 0.1$ is close to 1, consistent with the expectation that T_2^{XY8} is governed by the dipolar interaction between V_B^- , where the interaction strength $J \sim \rho \sim 1/r^3$. For the DROID sequence, the fitted $\alpha = 0.35 \pm 0.05$, exhibits a slower scaling with V_B^- density, coming from the cancellation of dipolar interaction. To estimate the V_B^- density from experiments, we minimize the relative squared residuals between the simulation and the experimentally obtained coherence timescales. To account for the uncertainty in estimating V_B^- concentration, we identify ρ that minimizes the residual as the best-fit parameter and estimate the error on this value as the range of ρ whose residuals lie within 5 %, shown in Figure R2. For the total boron-vacancy defect concentration obtained from SRIM simulation (e.g. line 171-172), we also include estimated error bars to account for the current fluctuation induced

uncertainties (measured to be $\sim 10\%$) in the ion implantation dosages.

We have included this new data analysis and the V_B^- density with estimated error bars into the main text and figures. We also include a detailed description of our density extraction procedure in the methods. We would like to thank the referee again for this really nice suggestion which helps us improve our manuscript.

FIG. R1: **Characterizing V_B^- density** Revised FIG. 3. from the main text. We have updated figure 3a with V_B^- densities extracted from the new procedure with associated error bars. We have also included an inset of adjusted fluorescence plotted against our estimated densities.

FIG. R2: **residuals** New added FIG. S3. from the methods. (a) Numerically simulated coherent timescales, \mathcal{T}_2 , for DROID and XY-8 pulse sequences. The solid lines show the timescales extracted from simulations with error bars plotted as semi-transparent color areas. Here we fit $\mathcal{T}_2 \propto \rho^{-\alpha}$. The sum of squared relative residuals of XY-8 and DROID between the experimental values and the fitted \mathcal{T}_2 plotted against V_B^- densities used in simulations. From bottom to top are residuals for sample S_1 , S_2 , and S_3 respectively, and the blue shaded regions are the $\pm 5\%$ error ranges from the minimum residuals.

3) Given the model presented by the authors, I find it confusing that the XY-8 T2 times appear to be roughly the same for S1 and S2, while the DROID T2 times appear to be about the same for S2 and S3 (at least within error bars based on Figure 1e and Figure 3a). To me this seems to at least partially invalidate the conclusion that V_B^- dipolar coupling is primarily responsible for decoherence under the XY-8 sequence, given the different implantation dosages and different extracted V_B^- densities across all the 3 samples. Do the authors have an explanation for this?

We thank the referee for raising this up, which helps us to better clarify and refine our results. First of all, our main evidence that the XY8 coherent timescale is limited by V_B^- dipolar coupling is that, for a given hBN sample, we always observe $T_2^D > T_2^{XY8}$. Compared to XY8 which only decouples the Ising interaction between V_B^- and the bath spins, DROID sequence further cancels the dipolar interaction within V_B^- themselves, thus leads to a improvement in measured coherence.

Secondly, after we re-analyze the estimated V_B^- density following the referee's previous suggestion in point (2), we find that the V_B^- concentration between sample S1 and S2 are close to each other within the error bar ($\rho_{V_B^-}^{S1} \approx 123_{-8}^{+7}$ ppm, $\rho_{V_B^-}^{S2} \approx 149_{-21}^{+25}$ ppm). Even for sample S3 ($\rho_{V_B^-}^{S3} \approx 236_{-31}^{+35}$ ppm), the estimated V_B^- density is only around 2-fold of sample S1's density. Therefore, the coherent timescales across all three samples are not expected to change significantly, albeit the difference in implantation dosages is 30-fold. This highlights another conclusion we want to make in this result that by increasing implantation dosage, one may not be able to create more V_B^- . We have revised the main text and added a short discussion in the methods to better clarify these two major conclusions.

4) The data set that seems to show the clearest trend with implantation dosage is actually T1, but the authors don't seem to address this at all. Is this consistent with their model?

We appreciate the referee's question and apologize for not commenting enough on the measured T_1 and T_1^p trends with different implantation dosages. One of the main reasons is that, in principle, dipolar flip-flop within V_B^- will not lead to

a decrease in T_1 . In the rotating frame, the dipolar interaction takes the form (main text Eq. 1):

$$\mathcal{H}_{\text{dip}} = \sum_{i < j} -\frac{J_0 \mathcal{A}_{i,j}}{r_{i,j}^3} (S_i^z S_j^z - S_i^x S_j^x - S_i^y S_j^y), \quad (\text{R1})$$

We can re-write the above interaction using raising and lowering operators

$$\mathcal{H}_{\text{dip}} = \sum_{i < j} -\frac{J_0 \mathcal{A}_{i,j}}{r_{i,j}^3} (S_i^z S_j^z - \frac{1}{2} [S_i^+ S_j^- + S_i^- S_j^+]). \quad (\text{R2})$$

Here, we can clearly see that dipolar interaction can lead to spin flip-flop between two nearby V_{B}^- ($|m_s = 0\rangle \otimes |m_s = -1\rangle \iff |m_s = -1\rangle \otimes |m_s = 0\rangle$). However, when measuring ensemble T_1 , we characterize dynamics of total spin polarization across the entire V_{B}^- ensemble, $\sum_i \langle S_i^z \rangle$, which remains unchanged under dipolar flip-flop. Therefore, T_1 is not expected to have a dependence on V_{B}^- concentration ρ .

In experiment, we have observed that T_1 indeed decreases with increasing ion dosages. This may be attributed to the presence of lattice damage during the implantation process or local charge state hopping. In fact, in the previous study of diamond NV centers with large electron irradiation dosages, people have observed significant shortening of NV T_1 , originating from ‘‘local spin fluctuators’’ that are associated with charge state hopping [1]. We have added a short discussion about the T_1 trend in both the main text and supplementary information to clarify this.

5) I would think the authors could use a combination of the measured fluorescence rates and the contrast differences between SB, SD, and SR to independently extract the V_{B}^- center densities in each sample and either validate or invalidate their extracted V_{B}^- densities. Is this analysis possible? If so, I feel it should be done. If not, why not?

We sincerely thank the referee for bringing this up. Following the referee’s suggestion, we went back to re-perform a systematic characterization of fluorescence counts (N) and contrast (C) across the three samples using the same laser power. The measured counts and contrasts are summarized in table R1.

Sample	S1	S2	S3
$\rho_{V_B^-}$ Extracted from Experiments (ppm)	123_{-8}^{+7}	149_{-21}^{+25}	236_{-31}^{+35}
Counts N ($\times 10^6$ photons/s)	0.64 ± 0.03	0.84 ± 0.03	1.44 ± 0.04
Contrast C (%)	4.74 ± 0.27	3.89 ± 0.19	4.00 ± 0.17
Adjusted Counts N_A (a.u.)	1.00 ± 0.06	1.07 ± 0.06	1.90 ± 0.11

TABLE R1: Summary of the estimated V_B^- concentration from coherent measurement, fluorescence counts, and contrasts for the three hBN samples

To account for the background from the measured fluorescence, we define the adjusted counts, N_A ,

$$N_A = N \times C. \quad (\text{R3})$$

and normalized it to 1 for sample S1. Interestingly, we find the adjusted fluorescence counts across three samples do follow the V_B^- densities estimated from spin coherent dynamics. We have included an inset in the new main text figure 3 [see Fig: R1 above] to highlight this agreement. We have also added a section in Methods to discuss this in more details. We thank the referee again for giving us the opportunity to validate our result.

6) When I first saw Figure 1a, I initially believed that experiments had been performed on V_B^- centers in monolayers of hBN, but after carefully reading the manuscript I believe this is not the case. I think Figure 1a should be modified to make it clear that these experiments are carried out in 100 nm thick hBN flakes with many layers, and that the dipolar interactions (and the spin bath) could include coupling to electronic and nuclear spins in other layers.

We thank the referee for this very good point and apologize for the confusion. Following the suggestion, we have made the corresponding adjustment to our main text Figure. 1a (see Figure. R3). Specifically, we have included two layers of lattice structures, as well as the schematic interactions between spins across the two layers. We also revise the caption to emphasize that the samples studied in this work have a thickness ~ 100 nm. We hope these changes could help to

FIG. R3: **Updated Fig 1a from the main text** Updated schematic of V_B^- spin ensemble (red spins) inside hBN crystal lattice (Nitrogen–blue; Boron–white); \hat{z} is defined along the c-axis (perpendicular to the lattice plane). \hat{x} and \hat{y} lie in the lattice plane, with \hat{x} oriented along one of the three V_B^- Nitrogen bonds. Two types of decoherence sources are presented here for V_B^- spin ensemble: the Ising coupling (grey wavy lines) to the bath spins (grey), and the dipolar interaction within V_B^- themselves (red wavy lines).

clarify our experiments and avoid potential confusion.

7) On line 128 the authors write “We fix the time intervals between pulses, $\tau_0 = 4\text{ns}$, sufficiently smaller than the correlation timescale of the local spin bath (estimated from the spin echo timescale). . .” How then is the XY-8 time varied in Figure 1d? Does this mean the number of pulses is increased for each subsequent XY-8 data point along the x axis, rather than increasing the free precession time between pi pulses? If so, I believe this is a fairly non-standard way of extracting T2 with dynamical decoupling sequences like XY-8, and this should also be stated explicitly and more clearly explained.

We thank the referee for the careful reading and apologize for not being clear in our previous manuscript. For both XY-8 and DROID measurement, we fix the time interval between pulses to be $\tau_0 = 4\text{ ns}$, and increase the number of pulses for each subsequent data point. There are two main reasons we are choosing this way of measuring coherence: (1) By fixing the time interval between pulses, the center frequency of the noise filter function of the applied sequence is then fixed during the measurement. As a result, we don’t need to worry about hitting

FIG. R4: T_2^{XY8} Comparison T_2^{XY8} measured on sample S3 with the highest ion implantation dosage using three different methods. The first two are to fix the XY-8 pulse number at $N_0 = 8$ and $N_0 = 16$ while increasing the pulse intervals. The third method is to fix the time interval between pulses to be $\tau_0 = 4$ ns while increasing the number of pulses, and this is the measurement technique we choose to use. Here dashed lines are data fitting with single exponential decays.

potential hyperfine coupling resonances between V_B^- and the nearby nuclear spin bath; and (2) By fixing $\tau_0 = 4$ ns and increasing pulse number, we are able to obtain enough data points at the early timescale to better capture the decoherence decay profiles.

For comparison, we have performed additional measurements of XY-8 on sample S₃ by fixing the pulse number at $N_0 = 8$ and $N_0 = 16$ while increasing the pulse intervals (see R4). We observe that for both cases, the XY-8 coherent timescale is shorter than the XY-8 measurement with a fixed pulse interval. This is not surprising as we expect the XY-8 sweep τ timescale should approach the XY-8 sweep N timescale at a large enough pulse number N_0 . However, at $N_0 = 16$, the first data point of the decay profile (corresponding to $\tau = 2$ ns) is already at 128 ns (including 16 π -pulses each with 6 ns duration), which is on the same order of the decay timescale one extract from the fitting.

We note that fixing pulse time interval τ_0 and sweeping pulse number N is also a commonly used measurement scheme to characterize spin coherence time T_2 and perform quantum sensing [2–8]. We have included a short discussion in the supplementary information about these two different ways of performing

XY-8 measurements and would like to thank the referee again for giving us the opportunities to clarify our results.

8) The authors write: “We note that the estimated d_{\perp} of V_{B}^{-} is on the same order of NV center in diamond. . .” Is this to be expected? Do the transverse electric field susceptibilities have similar origins in these two defects?

Very much like NV center in diamond, the ground state of V_{B}^{-} is an orbital singlet (and a spin triplet), leading to the naive expectation that a linear Stark shift is not allowed, thus the ground state of NV or V_{B}^{-} is not sensitive to the presence of external electric field at the leading order. For NV center in diamond, the measured small ground state transverse electric field susceptibility $d_{\perp}^{\text{NV}} \approx 17 \text{ Hz}/(\text{V}/\text{cm})$ has been attributed to the interplay between electric fields and the dipolar spin-spin interaction [9]. In particular, the effect is as follows: At first order in perturbation theory, the ground state wavefunction is mixed with the excited state by the presence of an electric field; this perturbation then couples to the ground-state spin degrees of freedom via the dipolar spin-spin interaction. This model successfully predicts an electric field susceptibility that is close to the measured value from experiment [10].

However, for V_{B}^{-} , to our best knowledge, there has been no clear experimental measurement or theoretical modeling of its ground state transverse electric field susceptibility. Our experiment provides an indirect estimation of d_{\perp} of V_{B}^{-} which appears to be on the same order of d_{\perp}^{NV} . This suggests that d_{\perp} of V_{B}^{-} may also be explained by the same origin as NV center. Indeed, since NV center and V_{B}^{-} have very similar ground state structures (orbital singlet and spin triplet), zero-field splittings (2.87 GHz and 3.48 GHz), optical zero-phonon lines (637 nm and 740 nm) and defect group symmetries (C_{3v} and D_{3h}), we would expect that the above-mentioned electric-field induced mixing between ground and excited states could result in a similar d_{\perp} between V_{B}^{-} and NV. However, we remark that this mechanism only offers a potential explanation for the measured d_{\perp} of V_{B}^{-} . Future experiments, such as directly measuring d_{\perp} by applying a large in-plane electric field, as well as theoretical studies based on DFT and group theory will provide more insights on this topic.

We would like to end by thanking the referee again for a tremendously careful reading of our manuscript and many helpful suggestions. A number of these suggestions have helped us significantly improve our manuscript, and we are particularly grateful.

Reviewer #2 (Remarks to the Author):

The Authors report on experimental measurements of an ensemble of interaction negatively charged boron vacancies in hexagonal boron nitride, a prominent two-dimensional material being investigated as a host for quantum defects. The Authors demonstrate a few results. They show that the coherence times for the measured samples are dominated by two effects, the coupling to the nuclear bath and the inter-defect dipolar interactions. This is shown by using two different decoupling sequences targeting the different interactions. From numerical simulations, the authors can estimate the percentage of defects created in the negative charge state from implantation. I find the results to be interesting and worthy of publication. I just have two minor questions to be addressed.

We thank the referee for a careful reading of our manuscript, insightful comments/questions and their strong support for publication.

1) “This is particularly prominent at high laser power, where the optical ionization of the defect charge state is enhanced.”

Calculations of the boron vacancy put the charge state transition levels far from the band edges (see e.g. <https://doi.org/10.1103/PhysRevB.97.214104>). Even a two-photon ionization process seems unlikely given the energetics. Are there any alternative processes that might explain the observations given the unlikely ionization?

We thank the referee for bringing this super relevant paper and apologize for not including this reference in our previous manuscript. We have taken a careful read of this paper, and it really helps better contextualize and understand our observation in the experiment. Notably, the calculated energy separation between the ground state of V_B^- and the conduction band of hBN is 4.46 eV. Therefore, a two-photon ionization process of V_B^- requires the energy of the pumping laser to be larger than 2.23 eV, or wavelength smaller than 556 nm. In our experiment, we apply a 532 nm laser with single photon energy 2.33 eV to excite V_B^- defects. Therefore in principle, the two-photon ionization process can still happen ($2 \times 2.33 \text{ eV} = 4.66 \text{ eV} > 4.46 \text{ eV}$). However, we totally agree with the

referee that, given this small energy difference, the two-photon ionization process is highly suppressed compared to NV centers in diamond, whose ground state is experimentally measured and theoretically calculated to be ~ 2.7 eV below the conduction band of diamond [11, 12]. This may actually explain the observation in our experiment that the ionization process of V_B^- only happens at large laser powers (~ 10 mW), while for previous studies of NV centers in diamond, people observe the ionization process at significantly lower laser power ($\sim 10 - 20$ μ W) [1, 11]. We have added a short discussion of this effect and included the relevant references in the revised main text to clarify this.

We note that unlike neutral NV^0 centers in diamond whose fluorescent emission is different from NV^- , neutral boron-vacancy (V_B^0) has not been directly observed from photo-luminescence signals. There may exist other potential processes and mechanisms than two-photon ionization that explain our observations, however future experimental and theoretical studies are required to understand this.

2) It is unclear to me whether the statement that “ $< 5\%$ of the created boron vacancy defects are in the desired negatively charged state” comes from simulations or experimental results. Given that there is some discrepancy between the numerical simulations and the exact experimental coherences that are measured, how reliable is the number? Perhaps the Authors can clarify this.

We thank the referee for pointing out this potential confusion and apologize for not being clear on this. To extract V_B^- charge state ratio $\eta = \rho_{V_B^-} / \rho_{V_B}$, we separately estimate $\rho_{V_B^-}$ from spin coherent measurement, and ρ_{V_B} from the ion implantation dosage and energy used to create defects. Specifically, for $\rho_{V_B^-}$, we fit the experimentally measured XY8 and DROID coherent timescales to theoretical simulations using 12 quantum spins, and estimate the V_B^- density from the dipolar interaction strength between spins. Following referee 1’s suggestion, we have re-performed the analysis of measurements and simulations to incorporate error bars of the estimated V_B^- densities (see the updated Fig. 3 R5). We attribute the discrepancy between experimental and simulated T_2 to pulse imperfections in the experiment, and finite-size effect, i.e. how many spins one can simulate, in numerical simulations.

For ρ_{V_B} , we employ SRIM (Stopping and Range of Ions in Matter), a widely used program for simulating ion implantation process, to estimate how many vacancies are created in the hBN samples from different helium implantation dosages and energies used in the experiment. We remark that at large implantation dosages (e.g. 10 ion/nm² in sample S3), the created vacancy density can be as high as $> 10^4$ ppm, leading to potential creations of vacancy clusters rather than individual vacancies. Therefore, our SRIM simulation may overestimate ρ_{V_B} at larger implantation dosage such as sample S3. However, this does not change the main conclusion in our work that the created negatively charged V_B^- does *not* scale linearly with the ion implantation dosage.

To be more precise in the revised manuscript, we have revised the statement as “a small portion of the created boron vacancy defects are in the desired negatively charged state”. We have also revised the manuscript to better clarify how we estimate the charge state ratio η .

FIG. R5: **Characterizing V_B^- density** Revised FIG. 3. from the main text. We have updated figure 3a with V_B^- densities extracted from the new procedure with associated error bars. We have also included an inset of adjusted fluorescence plotted against our estimated densities.

Again, we thank the referee for the endorsement of our manuscript and all the nice suggestions.

Reviewer #3 (Remarks to the Author):

In this paper the spin coherence properties of VB in hBN have been studied. Based on the decoherence mechanism of VB ensembles in these samples, the ratio of negatively charged VB to total boron vacancy defects has been modelled. At the end the electric field susceptibility of VB⁻ is calculated by comparing the splitting of E with difference density of VB. The decoherence mechanism has been reported in couple of works so far. The experiment has been done precisely and data have been analysed in details. The manuscripts could be organised better so the reader can follow the story. Currently the text jumps between figures which confuses the reader. Overall there are few important points that need clarification for this draft to be suitable for publication in a nature communication journal:

We thank the referee for the careful reading of our manuscript and apologize for the text jumping around between Figure 1 and Figure 2. The reason we chose to put the previous main text Figure 1d and 1e (coherent measurements) in front of Figure 2 (differential measurement scheme) was that we wanted to highlight the main results of the paper, i.e. the coherent dynamics of V_B^- under different pulse sequences, to the readers at the introduction, while leaving the detailed description of the differential measurement method in Figure 2. We were hoping that the readers could be immediately captivated by what is new in our result. However, we totally agree with the referee that it could lead to some confusion for the readers to follow the storyline. To this end, following the referee's suggestion, we have switched the previous main text Figure 1d and 1e with Figure 2, so that now the text and the figures are in the same order. We have also added reference to each figure in the introduction paragraph of the manuscript to present a clear road map of our results and the corresponding discussion around each figure at the very beginning. We thank the referee for this suggestion and hope that the updated manuscript now conveys a better-organized storyline to the readers.

The authors referred to charge state and optical ionization in the first part of manuscript and to explain the results in figure 2. Is there any experimental evidence on the charge state switching of negatively charged V_B^- ? Can the effects be explained considering initialisation

rate to spin 0 rather than charge state switching?

We appreciate the referee for highlighting this confusion. This point has been also echoed by referee 2 who has suggested a nice reference on the DFT calculation of the energies of different charge states of boron-vacancy defect [13]. In fact, from this paper, the calculated energy separation between the ground state of V_B^- and the conduction band of hBN is around 4.46 eV, slightly smaller than the energy of a two-photon process under 532 nm (2.33 eV) green laser excitation ($2 \times 2.33 \text{ eV} = 4.66 \text{ eV} > 4.46 \text{ eV}$). As a result, the two-photon ionization process from V_B^- to neutral V_B indeed can happen. However, given this small energy difference, the two-photon ionization process should be much weaker than NV centers in diamond, whose ground state is experimentally measured and theoretically calculated to be $\sim 2.7 \text{ eV}$ below the conduction band of diamond [11, 12]. This may actually explain the observation in our experiment that the ionization process of V_B^- only happens at large laser powers ($\sim 10 \text{ mW}$), while for previous studies of NV centers in diamond, people observe the ionization process at significantly lower laser power ($\sim 10 - 20 \mu\text{W}$) [1, 11]. However, unlike neutral NV^0 centers in diamond whose fluorescent emission is measured to be around 575 nm, neutral V_B^0 has not been directly observed from photo-luminescence signals. Therefore we also have to admit that the proposed ionization process in our manuscript only offers a potential explanation of the experiment. We have added a short paragraph after the ionization discussion in the main text to better clarify this. For the latter point: “can the effect be explained considering initialisation rate to spin 0 rather than charge state switching?”, we have performed additional measurements to rule out this possibility. In particular, if there exists a slow initialization of V_B^- to spin which causes the signal bright to rise at late times, then this spin polarization can be directly probed by performing an additional differential measurement before detection. The measurement sequence is shown in Figure R6: we first perform a spin echo sequence with fixed large time interval $\tau_1 = 600 \text{ ns} \gg T_2^{\text{Echo}}$, so that the system is no longer coherent and reaches a fully mixed state between $|m_s = 0, -1\rangle$, with measured contrast $C(t) \approx 0$. After that, we wait for a variable time τ_2 , and measure the

FIG. R6: **Revised Spin Echo Measurement** (a) Revised pulse sequence. Here we fix $\tau_1 = 600$ ns and sweep τ_2 . For S_D , an extra π -pulse is applied before the detection. (b) Comparison between normal Spin Echo and the revised measurement results at late times.

final contrast with an additional π -pulse in the signal dark sequence. We find that the difference between two signals, $C(t) = [S_B - S_D]/S_R$ remains around zero, indicating that no spin polarization is accumulated at late times. We hope this additional measurement could better clarify our proposed ionization process.

Authors mentioned that the discrepancy between the measured and simulated timescales may stem from imperfect spin rotations in the experiment, as well as finite-size effects from the simulations. Can they comment on why the simulation does not match to the experimental data. Even the relative rates are different in the simulations. Both experimental data show similar drop in the coherence time vs VB concentration but this is not the case in simulation result. In fact, the TXY8 drops very fast in the simulation and there is only very small change in the TDRIOD vs VB concentration.

We thank the referee for pointing this out, which is also raised by referee 1. To be honest, this discrepancy between simulation and experiment has bothered us a lot when preparing the manuscript. To better capture our experimental results, we attempted various models and employed a Krylov subspace exponentiation and eigensolving method in numerics to simulate the quantum dynamics of a dipolar system with $N = 12$ spins and many disorder averages, as well as including the finite pulse duration and the measured inhomogeneous broadening of V_B^- into our simulation, yet the simulation cannot completely reproduce the experiment results.

We believe the origins for the discrepancy are mainly pulse imperfections in the experiment and finite-size effects resulting from the limited number of spins one can simulate. Here we will explain in more details. First, the hyperfine interaction between V_B^- and nearby nuclear spins leads to a large broadening of the V_B^- transition with measured resonance line-width $\sigma \sim 80$ MHz (standard deviation). As a result, the Rabi driving frequency $\Omega \approx 83$ MHz that we can achieve in the experiment will lead to considerable imperfections since it is so close to the broadening σ . To account for this, a random static on-site field is applied on each simulated V_B^- spin with field strength drawn from the measured resonance distribution, and we average across all different configurations to reproduce the decoherence timescales. Nevertheless, the dynamics between bath spins is not included in the static on-site field, and the computation power restraint makes it difficult for us to fully capture this dynamical effect of a large number of spins in numerical simulations.

We note that this discrepancy between experiment and simulation is expected to be more pronounced in DROID sequence. As in XY8, the measured T_2 decay is governed by the dipolar-induced decoherence between V_B^- , while for DROID sequence where one decouples a large portion of the interactions, the pulse imperfection leads to a significant effect in the measured T_2 . In fact, if one could apply perfect control pulses, we would expect DROID T_2 to be nearly independent of V_B^- densities, as most of the interactions that lead to decoherence are canceled.

Therefore, we apologize for not being clear in our original manuscript and admit

that our model has some limitations. To this end, we have added a section in the updated methods section to include a detailed discussion on this discrepancy between the experiment and simulation. In addition, following a similar suggestion by Referee 1, we have softened our original claim in the abstract and main text that of “(we) directly determine the precise concentration of V_B^- ”, to now that “(we) directly estimate the concentration of V_B^- ”. Hope these changes better clarify the main discoveries in our manuscript.

It is important to provide V_B^- PL intensity VS implantation fluences and compare it to the result in this paper. Does it match with the proposed ratio of V_B^- upon implantation?

We sincerely thank the referee for bringing this really great suggestion into our attention (also suggested by Referee 1). Following the referees’ suggestion, we went back to re-perform a systematic characterization of fluorescence intensity (N) and V_B^- contrast (C) across the three samples at the same laser power (summarized in Table R2). To account for the potential different background fluorescence in three samples, we define the adjusted fluorescence counts, N_A ,

$$N_A = N \times C. \quad (\text{R4})$$

and normalized it to 1 for sample S1. Interestingly, we find the adjusted fluorescence counts across three samples do follow the V_B^- densities estimated from spin coherent dynamics. We have included an inset in the new main text Figure 3a [we also attached here in the referee report, see Fig: R7] to highlight this agreement. We have also added a section in Methods to discuss this in more details. We thank the referee again for giving us the opportunity to validate our results.

Sample	S1	S2	S3
$\rho_{V_B^-}$ Extracted from Experiments (ppm)	123_{-8}^{+7}	149_{-21}^{+25}	236_{-31}^{+35}
Counts N ($\times 10^6$ photons/s)	0.64 ± 0.03	0.84 ± 0.03	1.44 ± 0.04
Contrast C (%)	4.74 ± 0.27	3.89 ± 0.19	4.00 ± 0.17
Adjusted Counts N_A (a.u.)	1.00 ± 0.06	1.07 ± 0.06	1.90 ± 0.11

TABLE R2: Summary of the estimated V_B^- concentration from coherent measurement, fluorescence counts, and contrasts for the three hBN samples

FIG. R7: **Characterizing V_B^- density** Revised main text Figure 3. We have updated figure 3a with V_B^- densities extracted from the new procedure with associated error bars. We have also included an inset of adjusted fluorescence plotted against our estimated densities.

From Figure 4 it is clear that there is also zero field splitting shift in the ODMR vs fluences. Is there any effect of strain on the results in this paper? I suggest to add this discussion in the manuscript.

We thank the referee for pointing out the shift of zero-field splitting in the measured ODMR spectra with increasing dosages and apologize for not discussing it in the original manuscript. We totally agree that the implantation-induced strain effect can potentially explain the observed shifts. Indeed, several previous studies have shown that strain can result in a shift of zero-field splitting [14–17]. We add a short discussion with several associated references in the main text,

as well as a new extended figure (also attached here as Fig) in the methods part to discuss this effect.

FIG. R8: **ESR splittings and center frequencies** The ESR Splittings and Center Frequencies of three hBN samples extracted by fitting the experimental data to the sum of two Lorentzian functions.

Finally, we would like to the referee again for the very helpful comments and questions. We very much hope that our replies and revisions to the manuscript have helped to clarify and answer all of the referee's questions and that the referee will now be able to support our manuscript for publication in Nature Communications.

-
- [1] J. Choi, S. Choi, G. Kucsko, P. C. Maurer, B. J. Shields, H. Sumiya, S. Onoda, J. Isoya, E. Demler, F. Jelezko, et al., *Physical review letters* **118**, 093601 (2017).
 - [2] Z.-H. Wang, G. De Lange, D. Ristè, R. Hanson, and V. Dobrovitski, *Physical Review B* **85**, 155204 (2012).
 - [3] T. Van der Sar, Z. Wang, M. Blok, H. Bernien, T. Taminiiau, D. Toyli, D. Lidar, D. Awschalom, R. Hanson, and V. Dobrovitski, *Nature* **484**, 82 (2012).
 - [4] M. A. A. Ahmed, G. A. Alvarez, and D. Suter, *Physical Review A* **87**, 042309 (2013).

- [5] T. H. Taminiau, J. Cramer, T. van der Sar, V. V. Dobrovitski, and R. Hanson, *Nature nanotechnology* **9**, 171 (2014).
- [6] J. Choi, H. Zhou, H. S. Knowles, R. Landig, S. Choi, and M. D. Lukin, *Physical Review X* **10**, 031002 (2020).
- [7] H. Zhou, J. Choi, S. Choi, R. Landig, A. M. Douglas, J. Isoya, F. Jelezko, S. Onoda, H. Sumiya, P. Cappellaro, et al., *Physical review X* **10**, 031003 (2020).
- [8] C. Zu, F. Machado, B. Ye, S. Choi, B. Kobrin, T. Mittiga, S. Hsieh, P. Bhattacharyya, M. Markham, D. Twitchen, et al., *Nature* **597**, 45 (2021).
- [9] M. W. Doherty, J. Michl, F. Dolde, I. Jakobi, P. Neumann, N. B. Manson, and J. Wrachtrup, *New Journal of Physics* **16**, 063067 (2014).
- [10] M. Block, B. Kobrin, A. Jarmola, S. Hsieh, C. Zu, N. Figueroa, V. Acosta, J. Minguzzi, J. Maze, D. Budker, et al., *Physical Review Applied* **16**, 024024 (2021).
- [11] N. Aslam, G. Waldherr, P. Neumann, F. Jelezko, and J. Wrachtrup, *New Journal of Physics* **15**, 013064 (2013).
- [12] L. Razinkovas, M. Maciaszek, F. Reinhard, M. W. Doherty, and A. Alkauskas, *Physical Review B* **104**, 235301 (2021).
- [13] L. Weston, D. Wickramaratne, M. Mackoite, A. Alkauskas, and C. Van de Walle, *Physical Review B* **97**, 214104 (2018).
- [14] A. Gottscholl, M. Diez, V. Soltamov, C. Kasper, D. Krauße, A. Sperlich, M. Kianinia, C. Bradac, I. Aharonovich, and V. Dyakonov, *Nature communications* **12**, 4480 (2021).
- [15] T. Yang, N. Mendelson, C. Li, A. Gottscholl, J. Scott, M. Kianinia, V. Dyakonov, M. Toth, and I. Aharonovich, *Nanoscale* **14**, 5239 (2022).
- [16] X. Lyu, Q. Tan, L. Wu, C. Zhang, Z. Zhang, Z. Mu, J. Zúñiga-Pérez, H. Cai, and W. Gao, *Nano Letters* **22**, 6553 (2022).
- [17] D. Curie, J. T. Krogel, L. Cavar, A. Solanki, P. Upadhyaya, T. Li, Y.-Y. Pai, M. Chilcote, V. Iyer, A. Poretzky, et al., *ACS Applied Materials & Interfaces* **14**, 41361 (2022).

REVIEWERS' COMMENTS

Reviewer #1 (Remarks to the Author):

As discussed in my prior report, this manuscript, entitled “Coherent Dynamics of Strongly Interacting Electronic Spin Defects in Hexagonal Boron Nitride” by R. Gong et al. reports experimental results in which dynamical decoupling sequences are applied to ensembles of optically active negatively-charged boron-vacancy (VB-) centers in order to determine their ground state spin coherence times, the primary sources of decoherence, and the defect density in three samples implanted with different dosages. The primary new results are the use of several different pulse sequences and a differential measurement technique to study the role of dipolar interactions between VB- centers in decoherence and to estimate the VB- concentration, and the use of electron spin resonance at zero magnetic field to study the charge environment of the VB- centers and to estimate their transverse electric field susceptibility.

These results are of considerable interest to experts working with defects in 2D materials, and more broadly will be of some interest to anyone interested in solid-state quantum systems, and I believe they are appropriate for publication in Nature Communications. From my perspective the authors have thoroughly and convincingly addressed my questions and comments as well as those of the other two reviewers, and I agree with the authors' assessment that the quality of the manuscript has been improved as a result. I recommend publication in Nature Communications without further review.

Reviewer #2 (Remarks to the Author):

The Authors have adequately addressed my concerns, and I am positive towards publication in Nature Communications.

Reviewer #3 (Remarks to the Author):

The authors have been addressed all the comments in details and provided more data on VB PI intensity VS implantation dose that confirm the measured VB concentration. The revised manuscript

reads very well and all details are provided that clear all of my concerns. However, the main claim of the paper was on exact determination of VB- concentration by analyzing the spin dynamics in hBN. As mentioned by the authors they model cannot determine the exact concentration of VB- and only provides estimates. The estimates of VB concentration by analyzing spin dynamics has been reported before(i.e <https://www.science.org/doi/10.1126/sciadv.abf3630>). This would weaken the work to be suitable for publication in a journal with broad audiences such as nature communication. Hence I suggest the work to be submitted to a more focused journal.